# Frequency-Augmented Mixture-of-Heterogeneous-Experts Framework for Sequential Recommendation

## ABSTRACT

Recently, many efforts have been devoted to building effective sequential recommenders. Despite their effectiveness, these methods typically develop a single model to serve all users. However, our empirical studies reveal that different sequential encoders have intrinsic architectural biases and tend to focus on specific behavioral patterns, *i.e.,* particular frequency range of user behavior sequences. For example, the Self-Attention module is essentially a low-pass filter, focusing on low-frequency information while neglecting the high-frequency details. This evidently limits their ability to capture diverse user patterns, leading to suboptimal recommendations.

To tackle this problem, we present **FamouSRec**, a **F**requency-**A**ugmented **M**ixture-**o**f-Heterogene**ous**-Experts Framework for personalized **Rec**ommendations. Our approach builds an MoE-based recommender system, integrating the strengths of various experts to achieve diversified user modeling. For developing the MoE framework, as the key to our approach, we instantiate experts with various model architectures, aiming to leverage their inherent architectural biases and capture diverse behavioral patterns. For selecting appropriate experts to serve individuals, we introduce a frequency-augmented router. It first identifies frequency components in user behavior sequences that are suited for expert encoding, and then conducts customized routing based on the informativeness of these components. Building on this framework, we further propose two novel contrastive tasks to enhance expert specialization and alignment, thus improving modeling efficacy and enabling robust recommendations. Extensive experiments on five real-world datasets demonstrate the effectiveness of our approach. Code is available at: https://anonymous.4open.science/r/FamouSRec/.

## CCS CONCEPTS

• **Information systems → Recommender systems**.

## KEYWORDS

Sequential Recommendation, Mixture-of-Heterogeneous-Experts

**ACM Reference Format:**
Anonymous Author(s). 2018. Frequency-Augmented Mixture-of-Heterogeneous-Experts Framework for Sequential Recommendation. In *Proceedings of Make sure to enter the correct conference title from your rights confirmation emai (Conference acronym 'XX).* ACM, New York, NY, USA, 11 pages. https://doi.org/XXXXXXX.XXXXXXX

## 1 INTRODUCTION

Recommender systems have become prevalent to enhance user experience on various online platforms by predicting user potential interests from extensive item pools [7, 8]. As user behaviors naturally evolve over time, it is crucial to develop sequential recommendation (SR) methods to capture these dynamic features [11, 28, 34]. Early work employed Markov chains [26] to predict the next items. Recent advancements have largely improved recommendation performance, by integrating advanced deep learning architectures like RNNs [9], CNNs [31], and Transformers [11].

Though the adopted techniques are different, most existing methods follow the same paradigm: it builds a universal sequential model by fitting the behavior sequences of all users, aiming to capture the general relationships between user historical interactions and target items [11, 37]. The learned model is then applied uniformly to all users, predicting the most likely preferred item based on the observed sequential context. Consequently, the capacity of the globally shared model to capture each individual's behavior patterns is crucial for making personalized recommendations [34].

Despite the progress, existing studies indicate the inherent biases in model architectures [4, 21, 27]. For example, the self-attention module employed in Transformer essentially functions as a low-pass filter in the frequency domain, concentrating on low-frequency signals while neglecting high-frequency ones[1] [21, 24]. Therefore, we suspect that when making sequential recommendations, *different sequential encoders may be limited to specific frequencies of user behavior sequences, due to the architectural bias.* To examine this hypothesis, we conduct several empirical experiments in Section 2. We find that different models tend to focus on **particular** frequency components, lacking comprehensive modeling of diversified frequencies. In contrast, we also find that users exhibit **diverse** behavioral patterns[2]: some follow long-term preferences (low-frequency signals), while others are driven by high-frequency signals, making short-term interactions. These insights suggest that employing a single biased model to capture the diverse behavioral patterns of all users may not be optimal. Instead, it would be more effective to develop a universal recommendation framework that can employ the strengths of various sequential encoders personally.

Considering these issues, our solution is inspired by recent advances of *Mixture-of-Experts* (MoE). Typically, MoE is employed to solve the multi-task problem, by assigning each task to a specialized module or "expert" [1, 18]. Instead of focusing on multi-task solutions, we aim to employ MoE for fine-grained user modeling, activating customized experts tailored to *each individual*. This could allow specific experts to focus on particular user behavioral patterns at certain frequencies, providing more personalized sequential

---

[1]When analyzing the sequence data in the frequency domain, "low-frequency" signals typically capture an overall perspective of the sequence, whereas "high-frequency" signals reflect frequent interactions within a short period.
[2]In this paper, we use the frequency components within user behavior sequences as explicit representations of user behavioral patterns.

modeling. Furthermore, to leverage the intrinsic biases of model architectures, rather than using identical structures for all experts as in traditional MoE frameworks [10, 29], we consider instantiating distinct experts with **heterogeneous architectures**, thus perceiving a wider range of frequency information. Overall, such an MoE scheme can be denoted as "user behavior frequency ⇒ customized experts ⇒ captured behavioral patterns". While this approach is appealing, it presents several major challenges. First, as mentioned earlier, user behaviors exhibit diverse patterns and intertwined frequency components. It is unclear how to select appropriate experts to serve each user. Second, to achieve effective MoE modeling, it is crucial to improve expert specialization in capturing specific frequency ranges. Moreover, with these heterogeneous experts, we should also consider their alignments to ensure coherent recommendations and robust services.

To address these issues, in this paper, we propose the **F**requency-**A**ugmented **M**ixture-of-Heterogene**ous**-Experts Framework for Sequential **Rec**ommendations, named **FamouSRec**. Our approach aims to (1) develop an MoE framework to capture diverse user behavioral patterns, (2) activate customized experts for serving each individual. To achieve the first goal, our key innovation is designing a novel heterogeneous MoE framework that employs various model architectures, such as Self-Attention [33], GRU [9], Mamba [6], and MLP modules, as experts. This allows us to leverage the inherent biases of these models and capture a wider range of user behavioral patterns. For achieving customized routing, we introduce a frequency-augmented router. During recommendation serving, the router first identifies expert-focused frequency components within user behavior sequences and then activates the appropriate experts based on the informativeness of these components. With the above framework, we further propose expert *specialization* and *alignment*, to enhance modeling efficacy. To create specialized experts, we introduce an expert-frequency contrastive learning task that explicitly differentiates experts in the frequency domain. To achieve expert alignment for consistent recommendations, we design an expert alignment contrastive method, aligning each expert with the more powerful activated ones. Overall, our approach can activate tailored experts for each user and provide personalized recommendations.

To evaluate the proposed approach FamouSRec, we conduct extensive experiments on real-world datasets. The results demonstrate that our proposed approach can effectively router appropriate experts to infer user preferences and make personalized recommendations. The main contributions of this work are as follows:

• We identify the inherent bias of existing sequential models, and point out their inefficacy in capturing diverse behavior patterns.

• We build a heterogeneous MoE framework for sequential recommendation, which can capture diverse behavior patterns and provide customized recommendations.

• Extensive experiments have demonstrated the effectiveness of our approach in providing personalized recommendations.

## 2 EMPIRICAL STUDIES ON USER BEHAVIOR AND MODEL ARCHITECTURE

In this section, we employ signal processing methods to conduct empirical studies that evaluate (1) the variety of user behavior patterns in recommender systems, and (2) the capacities of distinct

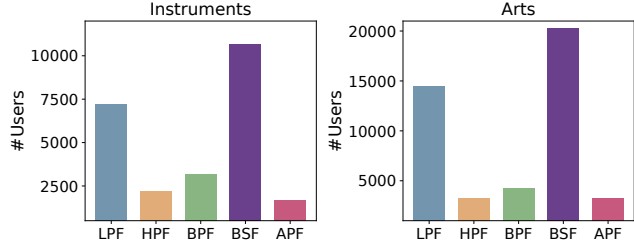

**Figure 1: The distribution of users who are driven by specific frequencies. We extract various frequency components on user behavior sequence, and calculate their relevance with ground-truth item, considering that user behaviors are driven by the most relevant components.**

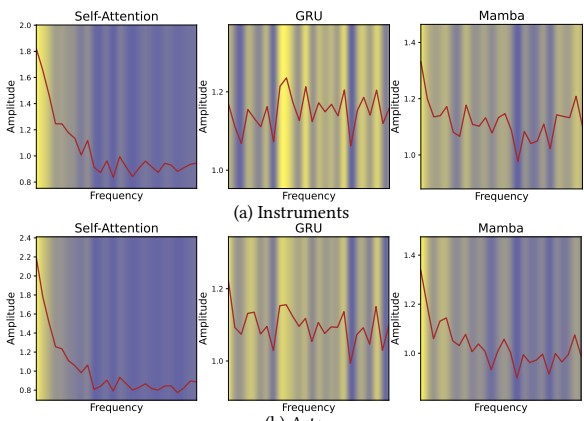

(a) Instruments

(b) Arts

**Figure 2: Visualization of frequency attention by different sequence encoder architectures.**

model architectures to capture these patterns. The experiments are conducted on the Amazon Instruments and Arts datasets [20].

**The variety of user behavioral patterns.** Here we explore whether different users exhibit diverse behavioral patterns. Considering that these patterns are typically intertwined in behavior sequences, it is non-trivial to clarify them in the time domain. Therefore, we shift our focus to the frequency domain and consider frequency components in interaction sequences as explicit representations of user behavior patterns. To achieve this, we first train a classical sequential recommender (*i.e.,* SASRec [11]), to obtain user behavior sequences representations. We then apply several signal processing algorithms, including low-pass filter (LPF, pass low-frequency signals while reducing high-frequency), high-pass filter (HPF), band-pass filter (BPF), band-stop filter (BSF), and all-pass filter (APF), to extract specific frequency components from these representations. Finally, we transform these frequency components back to the time domain and evaluate their relevance to the ground-truth items, hypothesizing that current user behaviors are primarily driven by the most relevant frequency components. We count how many users are driven by each frequency component. As illustrated in Figure 1, *users exhibit diverse behavioral patterns, each focusing on distinct frequencies*. This result highlights the importance of developing advanced models to capture personalized user behavior patterns.

**The architecture bias of existing sequential encoders.** In this part, we evaluate the efficacy of different sequential model architectures in capturing user diverse behavioral patterns. Especially, in this paper, we focus on several well-known encoder architectures in the literature of sequential recommendations, *i.e.,* Self-Attention, GRU, and Mamba. To be specific, inspired by [38], we add a learnable frequency filter layer between the item embedding layer and the sequence encoder layer (more details are described in Section 3.2.2). By training the hybrid model on user interaction data as usual, the frequency filter module can learn to extract frequency components best suitable for encoding by downstream sequence encoders. Figure 2 shows the learned frequency and amplitude of the filter model. As can be seen, *different architectures have inherent biases and demonstrate distinct frequency attentions.* Specifically, (1) for Self-Attention, the frequency filter learns to pass low-frequency signals and attenuate high-frequency ones, essentially serving as a low-pass filter. (2) GRU does not show a preference for specific frequency components, indicating its all-pass nature. (3) Mamba primarily focuses on low frequencies but also captures some high-frequency components, acting as a band-stop filter.

The above analysis motivates us to integrate the strengths of different architectures and select suitable experts for serving distinct users, thus providing customized recommendations. We further explore the efficacy of frequency domain learning in Appendix A.4.

## 3 METHODOLOGY

In this section, we present the details of the proposed **F**requency-**A**ugmented **M**ixture-of-Heterogene**ous**-Experts Framework for Sequential **Rec**ommendations, named **FamouSRec**. Our approach develops an MoE framework with heterogeneous experts, capable of routing suitable experts to encode diverse user behavior sequences. This enables the framework to capture user preferred behavioral patterns and provide personalized recommendations.

### 3.1 Approach Overview

The empirical findings in Section 2 have demonstrated that users tend to exhibit diverse behavioral patterns, which a single sequence encoder may struggle to capture due to its inherent architectural biases. To solve this, we develop the sequential recommender using an MoE framework as the backbone, activating specialized experts tailored for different users. Unlike previous studies that use the same network architecture for all experts, our key innovation is to instantiate experts with heterogeneous architectures, to capture distinct frequency components within user historical interactions. Specifically, to serve an individual user, a frequency-augmented router first analyzes the frequency components in user behavior sequences. It then activates a selected subset of experts to encode several informative components and provide personalized recommendations (Section 3.2). To further improve the modeling efficacy of the heterogeneous MoE framework and enable it to make coherent recommendations, we introduce two training strategies, namely expert-frequency contrastive learning (Section 3.3.1) and expert alignment contrastive learning (Section 3.3.2). Based on the above methods, the proposed framework can activate customized modules for each user, satisfying their personalized needs. The overall framework of FamouSRec is depicted in Figure 3.

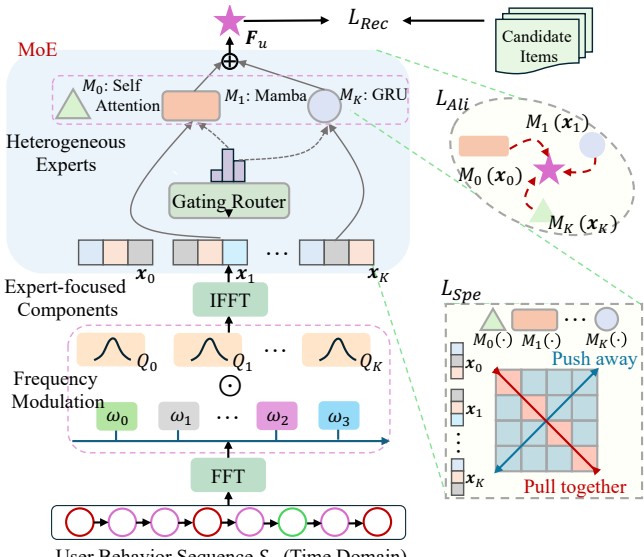

**Figure 3: The overall framework of our proposed FamouSRec.**

## 3.2 Frequency-Augmented Mixture-of-Heterogeneous-Experts Framework

To develop the MoE framework, we introduce the widely-used shared-bottom structure. This consists of a shared item embedding layer and $L$ MoE layers. Each MoE layer includes a frequency-augmented router and several heterogeneous experts to encode user diverse behavioral patterns. In what follows, we introduce these methods in detail.

*3.2.1 Embedding Layer.* In the embedding layer, we transform item identifiers into dense vectors using an item embedding matrix $V_I \in \mathbb{R}^{|I| \times d}$. For user behavior sequence $S_u = \{i_1, i_2, \cdots, i_n\}$, the look-up operation retrieves the corresponding embeddings, forming an input sequence representation $E \in \mathbb{R}^{n \times d}$. To incorporate positional information, a learnable positional encoding matrix $P \in \mathbb{R}^{n \times d}$ is added to the item embeddings. The embedding result is then processed through dropout and layer normalization to improve model stability and generalization. Thus, we can generate the sequence representation $E_u \in \mathbb{R}^{n \times d}$ of $S_u$ by:

$$E_u = \text{Dropout}(\text{LayerNorm}(E + P)). \tag{1}$$

*3.2.2 Frequency-Augmented Router.* The most crucial step in building the MoE framework is to activate appropriate experts for processing input data [1]. As our approach, considering that different model architectures tend to focus on distinct frequencies, we propose first extracting these "expert-focused" frequency components from user behavior sequences, and then deciding which experts to activate based on the informativeness of these components.

To implement our idea, given the user behavior sequence representation $H^l \in \mathbb{R}^{n \times d}$ of the $l$-th layer (we set $H^0 = E_u$, and we will omit the layer superscript for clarity), we first convert them into the frequency domain by performing Fast Fourier Transform (FFT, with details provided in Appendix A.1) along the item dimension:

$$\widetilde{H} = \mathcal{F}(H) \in \mathbb{C}^{n \times d}, \tag{2}$$

where $\mathcal{F}(\cdot)$ denotes the FFT operation and $\widetilde{H}$ is the spectrum of $H$ in the complex space.

Then, for each expert, we capture their focused frequency components from user behavior sequences. Especially, inspired by [38], we build $K$ learnable frequency filters $\{Q_1, \ldots, Q_K\}$, where $K$ is the total number of experts and $Q_j \in \mathbb{C}^{n \times d}$ is a complex tensor. By multiplying these filters with the spectrum of the input representations $\widetilde{H}$, we can modulate the spectrum and extract several expert-focused frequency components as follows:

$$\widetilde{H}'_j = Q_j \odot \widetilde{H}, \tag{3}$$

where $\odot$ is the element-wise multiplication, and $\widetilde{H}'_j$ represents the extracted frequency components suitable for $j$-th expert. Since these filters are learnable, we aim for them to be optimized to capture frequencies best suited for each expert to encode during training.

Finally, we assess expert significance by evaluating the informativeness of the extracted expert-focused frequency components. Specifically, we adopt the inverse FFT to transform these components $\widetilde{H}'_j$ back to the time domain. By concatenating these transformed results together and feeding them into a typical MLP module, we can obtain the gate-value for each expert $g_j$ as follows:

$$g = XW_1 + b_1, \tag{4}$$
$$X = [x_1; \ldots; x_K], \tag{5}$$
$$x_j = \mathcal{F}^{-1}(\widetilde{H}'_j). \tag{6}$$

where $\mathcal{F}^{-1}(\cdot)$ is the inverse FFT, $x_j$ is the time-domain representation of frequency component suitable for the $j$-th expert (in subsequent text, we abbreviate it as frequency components when the context is clear), $W_1 \in \mathbb{R}^{d \times K}$ and $b_1 \in \mathbb{R}^K$ are learnable parameters.

*3.2.3 Sparsely-Gated MoE.* To leverage the model architectural biases that focus on distinct frequencies, in this work, we instantiate heterogeneous experts with several classic architectures, including self-attention, GRU, MLP, and Mamba encoders. This could naturally enhance the specialization of expert capabilities, thus improving the efficacy of the MoE framework [10]. Especially, the extracted $K$ expert-focused frequency components (see Eqn. (3) and Eqn.(6)) are used as inputs for these experts to encode, with each component uniquely corresponding to an expert. Additionally, existing MoE-based recommender systems [17, 18] typically aggregate results from all experts to make inferences. Despite the efficacy, this method activates all experts during each forward pass, resulting in high computational costs. To enhance inference efficiency, our approach employs a sparsely-gated MoE framework during serving [1]. Formally, given the gate-value for each expert $g$, we activate the most appropriate $k$ experts to encode their focused frequency components within user behavior sequences, and combine their results to infer user potential interests as follows:

$$F_u = \sum_{j=1}^{K} g'_j \cdot M_j(x_j), \tag{7}$$
$$g' = \text{Softmax}(\text{Top-k}(g, k)), \tag{8}$$
$$\text{Top-k}(g, k)_j = \begin{cases} g_j & \text{if } g_j \text{ is in the top-}k \text{ elements of } g, \\ -\infty & \text{otherwise.} \end{cases} \tag{9}$$

where $M_j(\cdot)$ represents the $j$-th expert encoder, and $x_j$ is the time-domain representation of the frequency components that the $j$-th expert should focus on (as defined in Eqn. (6)), $F_u$ denotes the encoded hidden representation of user $u$ behavior sequence. We take the final hidden vector $f_n$ corresponding to the $n$-th (last) position as the sequence representation.

*3.2.4 Prediction Layer.* Given the final representation of user behavior sequences $f_n$ (see Eqn. (7)), we can finally predict the next potential item according to the following probability:

$$P(i_{n+1}|s) = \text{Softamx}(f_n \cdot e_{i_{t+1}}). \tag{10}$$

where we calculate the softmax probability over the candidate item set, and $e_{i_{t+1}}$ is the embedding representation of item $i_{t+1}$ from embedding matrix $V_I$.

## 3.3 Expert Specialization and Alignment

By training the above frequency-augmented MoE framework to fit user behavior data, we aim to achieve expert specialization, where each expert focuses on non-overlapping and particular frequency components. This could reduce knowledge redundancy and enhance the modeling efficiency of the MoE framework in capturing diverse user behavioral patterns [1, 10]. However, due to the limited number of experts and the variety of frequency components in user behavior sequences, simply relying on this implicit training paradigm may not effectively distinguish the intertwined frequencies [4]. As our solution, we introduce an expert-frequency contrastive learning task, aiming to specialize the experts in the frequency domain explicitly. Furthermore, while it is essential to enhance the expert specialization, due to the heterogeneous nature of these experts, we should also consider expert alignment to make consistent recommendations. Therefore, we further propose an expert alignment contrastive task, aiming to improve the alignments between different experts and the more powerful activated experts. In the following sections, we introduce these methods in detail.

*3.3.1 Expert-Frequency Contrastive Learning.* In this task, we aim is to explicitly guide each expert to focus on specific frequency components within user behavior sequences, thus improving expert specialization. To achieve this goal, we propose contrasting the prediction results of experts when encoding different frequency components. This involves aligning each expert with its corresponding frequency components while deliberately misaligning them with frequencies that other experts should focus on.

To be specific, given the incorporated experts $\{M_1, \ldots, M_K\}$ and their corresponding expert-focused frequency components $\{x_1, \ldots, x_K\}$, we first employ the experts to encode these different frequency components, resulting in $K \times K$ pairs. Then we use these encoded results to generate item prediction logits. For all prediction logits from matched pairs (i.e., $M_j(x_j)$), we consider them as positive samples and aim to minimize their Kullback-Leibler divergence. Conversely, logits from mismatched pairs of experts and frequency components (i.e., $M_j(x_{j', j' \neq j})$) are taken as negative samples and pushed away. The expert-frequency contrastive loss can be formalized as follows:

$$\mathcal{L}_{\text{Spe}} = \frac{\sum_{j=1}^{K} \sum_{k=1, k \neq j}^{K} \text{KL}(p_{j,j} \parallel p_{k,k})}{\sum_{(j,k) \in \mathbb{P}} \sum_{\substack{(j',k') \in \mathbb{P} \\ (j,k) \neq (j',k')}} \text{KL}\left(p_{j,k} \parallel p_{j',k'}\right)}, \tag{11}$$

$$\text{KL}(p_{j,k} \parallel p_{j',k'}) = \sum_{i=1}^{|\mathcal{I}|} p_{j,k}(i) \log \frac{p_{j,k}(i)}{p_{j',k'}(i)}, \tag{12}$$

$$p_{j,k}(i) = \frac{\exp(M_j(\boldsymbol{x}_k) \cdot \boldsymbol{v}_i)}{\sum_{i'=1}^{|\mathcal{I}|} \exp(M_j(\boldsymbol{x}_k) \cdot \boldsymbol{v}'_i)}, \tag{13}$$

where $\mathbb{P} = \{(j,k) | 1 \leq j \leq K, 1 \leq k \leq K\}$ denotes all pairs of distinct experts and frequency components, and $p_{j,k}$ is the logits predicted by $k$-th expert when encoding $j$-th frequency component.

*3.3.2  Expert Alignment Contrastive Learning.* In addition to conducting expert specialization, we further consider expert alignment to make coherent recommendations. To achieve this, we propose the expert alignment contrastive task. Specifically, given the encoding result of each expert, we employ the weighted output of the MoE framework as positive samples and align them in the semantic space, while taking in-batch data as negative samples. The expert alignment contrastive loss can be formally presented as follows:

$$\mathcal{L}_{\text{Ali}} = -\sum_{i=1}^{B} \sum_{j=1}^{K} \log \frac{\exp(M_j(\boldsymbol{x}_j) \cdot f_i/\tau)}{\sum_{i'=1}^{B} \exp(M_j(\boldsymbol{x}_j) \cdot f_{i'}/\tau)}, \tag{14}$$

where $f_i$ is the weighted output of the MoE framework (see Eqn. (7)), and $f_{i'}$ is the negative samples encoded from in-batch data.

## 3.4  Optimization and Inference

Given the encoded representation of user behavior sequences $f_n$ (Eqn. (7)), we adopt the widely-used cross-entropy loss to optimize the recommendation capacity as follows:

$$\mathcal{L}_{\text{Rec}} = -\log \frac{\exp(f_n \cdot \boldsymbol{e}_i/\tau)}{\sum_{i'=1}^{|\mathcal{I}|} (\exp(f_n \cdot \boldsymbol{e}'_i/\tau))}, \tag{15}$$

where $\tau$ is the temperature parameter.

Notably, existing studies indicate that the sparsely-gated MoE framework may suffer from load balancing issues, where some experts are frequently activated while others are rarely used [1, 5]. This could reduce model efficiency. To address this problem, we introduce an auxiliary load balancing loss to promote a more even distribution of expert activations, as follows:

$$\mathcal{L}_{\text{Bal}} = \left(\frac{1}{B} \left(\sum_{u=1}^{B} \boldsymbol{g}_u\right) - \frac{1}{K}\right)^2, \tag{16}$$

where $\boldsymbol{g}_u$ is the gate value for serving the $u$-th user (Eqn. (4)), $B$ is the batch size, and $K$ is the total number of experts. The learned even distribution can also improve the expert specialization implicitly.

Overall, to jointly optimize the framework, we combine the loss functions from the recommendation task, the expert-frequency contrastive task, the expert alignment contrastive task, and the load balancing loss as follows:

$$\mathcal{L} = \mathcal{L}_{\text{Rec}} + \lambda \mathcal{L}_{\text{Spe}} + \mu \mathcal{L}_{\text{Ali}} + \gamma \mathcal{L}_{\text{Bal}}. \tag{17}$$

where $\lambda$, $\mu$, and $\gamma$ are weight hyper-paratmers. Notably, we can activate suitable experts to make inferences using Eqn. (10).

## 3.5  Discussion

In this section, we compare related methods to highlight the novelty of our approach. We analyze the time complexity in Appendix A.2 and indicate the efficiency of sparsely-gated MoE framework.

**General sequential methods** such as GRU4Rec [9] and SAS-Rec [11] adopts advanced architecture to encode user behavior sequences. However, as illustrated in Section 2, these models demonstrate inherent biases and tend to focus on particular frequencies. Therefore, this limits the efficacy of these single model based methods in capturing user diverse behavioral patterns, leading to sub-optimal results. As a comparison, our approach develops an MoE framework that can leverage strengths from various experts to achieve diverse encoding and provide customized service.

**Frequency augmented methods** such as FMLP-Rec [38] and FEARec [4] propose to enhance sequential modeling by incorporating frequency domain features. FMLP-Rec introduces a global filter to extract signals, focusing on a limited frequency range. FEARec alleviates this by introducing a frequency ramp structure that captures a wider range of frequencies. Nevertheless, these methods neglect to explore the relations between frequency components and encoders. For our approach, we extract specific frequencies tailored for different experts and specialize the experts to focus on these particular frequencies, thereby enhancing the modeling efficacy.

**Aggregation sequential methods** [15, 18] typically use techniques like MoE or model ensemble to integrate the strengths of various modules. However, most existing methods focus on *multi-task* or *cross-domain* problems. Notably, we focus on *fine-grained user modeling* and employ the MoE framework to provide customized recommendations, activating distinct experts tailored to each individual. Furthermore, most existing methods overlook the benefits of combining heterogeneous modules. Although a recent study M3 [30] develops the MoE framework with distinct expert architectures, it lacks explicit analysis of model inherent capacities and simply trains the model in an implicit way, leading to potential expert conflicts and redundancy. In contrast, our approach explicitly analyzes the expert architecture bias and develops a novel heterogeneous MoE, leveraging their inherent biases to capture diverse user behavioral patterns. Two contrastive tasks on expert specialization and alignment are proposed to enhance modeling efficiency and enable coherent recommendations.

## 4  EXPERIMENT

## 4.1  Experiment Setup

*4.1.1  Dataset.* To evaluate the performance of the proposed approach, we conduct experiments on five open public benchmark datasets: (1) **Instruments, Arts, and Office**: these three datasets are from Amazon review datasets in [20]. We select three subcategories: "Musical Instruments", "Arts, Crafts and Sewing", and "Office Products". (2) **Online retail (OR)** [2] contains transaction records from an e-commerce platform in the UK. (3) **Tmall** [32] is from the IJCAI-15 competition, and contains user shopping logs on Tmall online platform. Following previous work [37], we keep the five-core datasets and filter users and items with fewer than five interactions. Then we group the interactions by users and sort

**Table 1: Statistics of the preprocessed datasets. "Avg.$n$" is the average length of behavioral sequences.**

| Datasets | #Sequences | #Items | #Actions | Avg.$n$ | Sparsity |
|---|---|---|---|---|---|
| Instrument | 24,962 | 9,964 | 183,964 | 7.37 | 99.93% |
| Arts | 45,486 | 21,019 | 349,664 | 7.68 | 99.96% |
| Office | 87,346 | 25,986 | 597,491 | 6.84 | 99.97% |
| OR | 16,520 | 3,469 | 503,386 | 30.47 | 99.12% |
| Tmall | 66,909 | 37,367 | 427,797 | 6.39 | 99.98% |

them by timestamp ascendingly. The maximum sequence length is set to 50. The statistics of datasets are summarized in Table 2.

*4.1.2 Baseline Models.* We compare our proposed approach with several representative types of sequential recommendation models:

• **General sequential methods** employs advanced sequential models to encode user historical interactions. We consider the following methods as baselines: (1) **SASRec** [11] is a classic transformer-encoder based sequential recommendation model, which employs a multi-head self-attention mechanism to capture sequential patterns. (2) **BERT4Rec** [28] is a bidirectional self-attention recommender that employs a cloze prediction task to enhance the sequence encoding. (3) **GRU4Rec** [9] applies the classic GRU module to encode user behaviors. (4) **Mamba4Rec** [16] employs selective SSMs to achieve efficient sequential modeling.

• **Contrastive learning sequential methods** are built upon general models and aim to enhance sequential encoding by applying data augmentation for contrastive learning: (5) **CL4SRec** [35] proposes three data augmentation approaches to construct self-supervision signals. (6) **DuoRec** [23] introduces both unsupervised and supervised sampling strategies for contrastive learning.

• **Frequency augmented sequential methods** analyze user behavioral patterns in the frequency domain, further enhancing time-domain modeling. We consider the following methods as baselines: (7) **FMLP-Rec** [38] uses learnable frequency filters to denoise user behavioral sequences and extract valuable features. (8) **FEARec** [4] enhances time domain attention and learns both low-high frequency information with a ramp structure.

• **Aggregation sequential methods** enhance recommendation performance by integrating the strengths of various sub-modules: (9) **Crocodile** [15] introduces a mixture-of-embedding-experts framework, and employs a covariance loss to disentangle different experts, thereby capturing user diverse interests. Notably, Crocodile is initially designed for cross-domain settings. But for a fair comparison, we reimplement it by instantiating multiple item embeddings within a single domain and inputting them for sequential encoders. (10) **EMKD** [3] predicts next items by averaging the encoding results of multiple encoders, and employs knowledge distillation to facilitate knowledge transfer between these encoders.

*4.1.3 Evaluation Settings.* To evaluate the performance of the next item prediction task, we adopt two widely used metrics, *i.e.,* HR@$N$ and NDCG@$N$, where $N$ is set to 5 and 10. Consistent with previous works [37], we apply the leave-one-out strategy for evaluation. For each user/session interaction sequence, the last item serves as the test data, the second-to-last item is used for validation, and the remaining interaction records are used for training. For evaluation,

we rank the target item of each sequence against all other items in the test set, and report the average score across all test samples.

*4.1.4 Implementation Details.* We implement the proposed FamouSRec model and all the compared baseline methods in PyTorch. To ensure a fair comparison, we optimize our model and the baseline methods with Adam optimizer and conduct a thorough hyperparameter search. We examine the effects of key hyper-parameters in Appendix A.3. We tune the number of experts in $\{3, 4\}$ and adjust the combination of different expert architectures. During training, we activate all the expert to optimize the proposed contrastive tasks. Notably, when making inferences, we adopt the top-$k$ sparsely-gated MoE framework, and set $k = 1$ to activate the best appropriate expert for serving. This further enhances the inference efficiency. The batch size is set to 4,096. All the experiments are conducted on a NVIDIA A100 GPU. We use the early stopping strategy with a patience of 10 epochs to prevent overfitting, and NDCG@10 is used as the indicator metric.

## 4.2 Overall Performance

We compare the proposed approach *FamouSRec* with baseline methods on the five target datasets. The results are reported in Table 2. . In general, FamouSRec outperforms baselines on nearly all datasets, leading to an average improvement ratio of 8%.

First, traditional general sequential recommendation methods such as SASRec, BERT4Rec, and GRU4Rec do not perform well, possibly because their architectural biases limit them to focusing on specific frequencies of user behavior sequences, making them challenging to capture diverse user behavior patterns. Notably, Mamba4Rec achieves impressive results, highlighting the effectiveness of state space models in modeling sequential dependencies [6]. Furthermore, as discussed in Section 2, the Mamaba module can function as a band-stop filter, which aligns with the behavioral patterns of most users. Additionally, contrastive learning methods such as CL4SRec and DuoRec outperform general methods. This suggests the effectiveness of contrastive learning in capturing latent semantic relations. Frequency augmented sequential methods (*i.e.,* FMLP-Rec and FEARec) demonstrate remarkable recommendation performance, highlighting the significance of frequency domain modeling. However, they neglect to extract particular frequencies suited for encoding by downstream modules, leading to sub-optimal results. Additionally, aggregation sequential methods Crocodile and EMKD do not yield satisfactory results, as they mainly combine models with the same architecture, leading to homogeneous capacity and diminishing the benefits of model aggregations.

Finally, our proposed model, FamouSRec, consistently outperforms all baseline methods across nearly all scenarios. Different from these baselines, we develop a mixture-of-hetergenous-experts framework to encode diverse user behavioral patterns. Leveraging a sparsely-gated architecture, our approach achieves superior performance while **activating a comparable number of parameters**. Notably, our method allows specific experts to focus on user behavior patterns at certain frequencies. By combining the strengths of these experts, our approach captures a broader range of user behaviors and offers personalized services. This remarkable result aligns with the success of the MoE framework in the field of NLP [5, 10].

**Table 2: Overall performance of different recommendation methods. The best and the second-best performance methods are denoted in bold and underlined fonts, respectively. "Improv." denotes the relative improvement ratios of the proposed approach over the best performance baselines.**

| Dataset | Metric | SASRec | BERT4Rec | GRU4Rec | Mamba4Rec | CL4SRec | DuoRec | FMLP-Rec | FEARec | Crocodile | EMKD | FamouSRec | Improv. |
|---|---|---|---|---|---|---|---|---|---|---|---|---|---|
| Instruments | HR@5 | 0.0851 | 0.0680 | 0.0728 | 0.0778 | 0.0791 | 0.0879 | 0.0834 | 0.0861 | 0.0839 | 0.0758 | **0.0946** | +7.62% |
| | NDCG@5 | 0.0571 | 0.0523 | 0.0590 | 0.0649 | 0.0530 | 0.0614 | 0.0564 | 0.0576 | 0.0597 | 0.0575 | **0.0794** | +22.34% |
| | HR@10 | 0.1102 | 0.0853 | 0.0915 | 0.0963 | 0.1045 | 0.1146 | 0.1081 | 0.1106 | 0.1082 | 0.0986 | **0.1180** | +2.97% |
| | NDCG@10 | 0.0652 | 0.0578 | 0.0650 | 0.0708 | 0.0612 | 0.0712 | 0.0644 | 0.0655 | 0.0674 | 0.0648 | **0.0869** | +22.05% |
| Arts | HR@5 | 0.0804 | 0.0510 | 0.0570 | 0.0714 | 0.0701 | 0.0803 | 0.0810 | 0.0802 | 0.0784 | 0.0727 | **0.0846** | +4.44% |
| | NDCG@5 | 0.0533 | 0.0376 | 0.0439 | 0.0582 | 0.0487 | 0.0557 | 0.0529 | 0.0536 | 0.0529 | 0.0567 | **0.0670** | +15.12% |
| | HR@10 | 0.1053 | 0.0671 | 0.0734 | 0.0896 | 0.0945 | 0.1048 | 0.1073 | 0.1059 | 0.1042 | 0.0949 | **0.1103** | + 2.79% |
| | NDCG@10 | 0.0613 | 0.0428 | 0.0492 | 0.0641 | 0.0566 | 0.0636 | 0.0613 | 0.0618 | 0.0604 | 0.0639 | **0.0751** | +17.16% |
| Office | HR@5 | 0.0913 | 0.0680 | 0.0774 | 0.0919 | 0.0826 | 0.0929 | 0.0926 | 0.0964 | 0.0924 | 0.0774 | **0.1020** | +5.81% |
| | NDCG@5 | 0.0674 | 0.0560 | 0.0658 | 0.0791 | 0.0609 | 0.0726 | 0.0683 | 0.0699 | 0.0688 | 0.0649 | **0.0855** | +8.09% |
| | HR@10 | 0.1109 | 0.0806 | 0.0908 | 0.1067 | 0.1007 | 0.1124 | 0.1136 | 0.1188 | 0.1118 | 0.0922 | **0.1217** | +2.44% |
| | NDCG@10 | 0.0738 | 0.0600 | 0.0701 | 0.0838 | 0.0667 | 0.0789 | 0.0751 | 0.0771 | 0.0756 | 0.0697 | **0.0916** | +9.31% |
| OR | HR@5 | 0.0815 | 0.0798 | 0.0822 | 0.0825 | 0.0546 | 0.0784 | 0.0798 | 0.0794 | 0.0782 | 0.0444 | **0.0852** | +3.27% |
| | NDCG@5 | 0.0492 | 0.0486 | 0.0496 | 0.0508 | 0.0342 | 0.0470 | 0.0479 | 0.0479 | 0.0474 | 0.0279 | **0.0524** | +3.15% |
| | HR@10 | 0.1402 | 0.1390 | 0.1418 | 0.1421 | 0.0955 | 0.1380 | 0.1391 | 0.1364 | 0.1384 | 0.0760 | **0.1493** | +5.07% |
| | NDCG@10 | 0.0680 | 0.0676 | 0.0696 | 0.0701 | 0.0473 | 0.0661 | 0.0670 | 0.0663 | 0.0662 | 0.0381 | **0.0730** | +4.14% |
| Tmall | HR@5 | 0.2721 | 0.1599 | 0.1917 | 0.2887 | 0.2274 | 0.2783 | 0.2906 | 0.2918 | 0.2784 | 0.2234 | **0.3011** | +3.19% |
| | NDCG@5 | 0.2425 | 0.1381 | 0.1671 | 0.2500 | 0.2017 | 0.2492 | 0.2505 | 0.2513 | 0.2516 | 0.1934 | **0.2591** | +2.98% |
| | HR@10 | 0.3013 | 0.1781 | 0.2135 | 0.3204 | 0.2496 | 0.3084 | 0.3197 | 0.3219 | 0.3089 | 0.2513 | **0.3395** | +5.47% |
| | NDCG@10 | 0.2519 | 0.1440 | 0.1741 | 0.2606 | 0.2134 | 0.2558 | 0.2604 | 0.2622 | 0.2583 | 0.2025 | **0.2705** | +3.17% |

**Table 3: Ablation analysis on two datasets. The best performance is denoted in bold.**

| Variants | Instruments | | Arts | |
|---|---|---|---|---|
| | HR@10 | NDCG@10 | HR@10 | NDCG@10 |
| FamouSRec | **0.1180** | **0.0869** | **0.1103** | **0.0751** |
| *w/o* Router | 0.1128 | 0.0804 | 0.1028 | 0.0714 |
| *w/o* Frequency Domain Learning | 0.1025 | 0.0759 | 0.0941 | 0.0637 |
| *w/o* Load Balance | 0.1074 | 0.0788 | 0.1075 | 0.0735 |
| *w/o* Expert Specialization | 0.1112 | 0.0813 | 0.1012 | 0.0703 |
| *w/o* Expert Alignment | 0.1119 | 0.0816 | 0.1038 | 0.0690 |

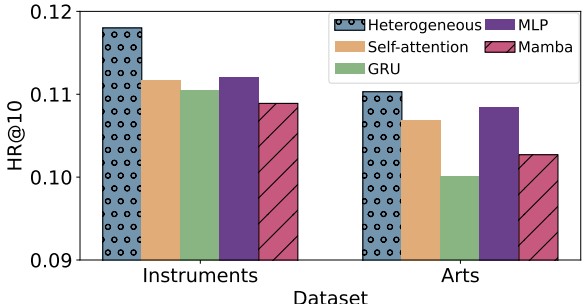

**Figure 4: Performance Comparison w.r.t. using experts with different architectures. "Heterogeneous" refers to our approach of using distinct architectures, whereas other variants use homogeneous experts to build the MoE framework.**

## 4.3 Further Analysis

*4.3.1 Ablation Study.* In this part, we analyze how each of the proposed components affects final recommendation performance. Table 3 shows the performance of our default method and its five variants on two datasets. Overall, removing any component degrades performance:

(1) *w/o* Router: In this variant, we simply average the output of different experts. The performance drop indicates that the router is effective at activating the most appropriate experts to encode sequential context, thus enabling customized inference.

(2) *w/o* Frequency Domain Learning: Instead of extracting specific frequency components for the router and experts to model, here we replace Eqn. (2) and Eqn. (3) with an MLP module and directly conduct time-domain modeling. The performance gap highlights the significance of frequency domain learning, which enables experts to focus on their suited frequency information for encoding.

(3) *w/o* Load Balance: Here, we exclude the load balancing loss, *i.e.,* Eqn. (16). This omission not only leads to a drop in performance but also makes the training process highly unstable, with some experts never being activated and trained. This highlights the effects of evenly distributing expert activations.

(4) *w/o* Expert Specialization: The performance drops sharply without the expert-frequency contrastive task (Eqn. (11)). This indicates that simply fitting user behavior could not effectively specialize heterogeneous experts in encoding specific frequencies.

(5) *w/o* Expert Alignment: In this variant, we omit the expert alignment contrastive loss, *i.e.,* Eqn. (14). The performance drop indicates that it is necessary to align heterogeneous experts for providing coherent recommendations.

*4.3.2 Effects of Expert Heterogeneity.* To evaluate whether incorporating heterogeneous experts can model a wider range of frequencies and capture diverse user behavioral patterns, we compare the results of instantiating experts with different architectures. Figure 4 shows that building the MoE framework by using experts with the same architectures results in lower performance than our heterogeneous MoE framework. Although these variants can learn to encode specific patterns for distinct experts implicitly, they neglect the effect of inherent architectural biases, restricting these experts to

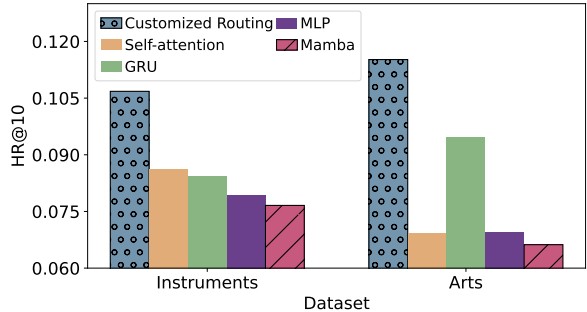

**Figure 5: Performance Comparison w.r.t. activating different experts. "Customized Routing" employs the proposed router to dynamically select the most appropriate expert for each individual, while other variants activate fixed experts.**

specific patterns. In contrast, our heterogeneous expert-based MoE framework effectively captures a wider range of user behavioral patterns, enhancing personalized user modeling.

*4.3.3 Effects of Customized Routing.* In this part, we evaluate whether the proposed MoE framework can select the most suitable experts to provide customized recommendations, given varying user sequential contexts. To verify this, we compare the recommendation performance when the router activates a tailored expert against when it consistently activates a fixed expert (we train the MoE framework as usual). As illustrated in Table 5, regardless of which expert is consistently activated, the performance significantly lags behind that of customized activation with the learned router. This result indicates that our method can build highly specialized experts for different patterns and can effectively route appropriate experts for each user, encoding their preferred patterns.

## 5 RELATED WORK

**Sequential recommendation.** Due to the dynamic nature of user preference, researchers propose developing sequential recommendation methods to predict future items by analyzing hidden patterns in user behavior sequences. Early works relied on the Markov Chain assumption [19, 26], estimating item-item transition probability matrices to predict the next item. With advances in deep learning, deep neural networks have become popular for sequential recommendation [9, 11, 37]. Numerous works based on Recurrent Neural Networks (RNNs) [13, 25], Convolutional Neural Networks (CNNs) [31, 36], and Transformers [11, 28] have been proposed to perform sequential recommendations. Recently, some advanced architectures, such as RWKV [22] and Mamba [6], have been proposed to enhance the efficiency of sequence modeling. There is also an increased emphasis on incorporating frequency domain features [27, 38]. Specifically, FMLP-Rec [38] introduces a learnable frequency filter to capture frequency components in user behavior sequences, building a filter-enhanced MLP architecture. FEARec [4] improves on this by using a frequency ramp structure to capture diverse frequency components for better modeling. Despite the efficacy, most of them overlook the relationship between model architectures and their encoded frequencies. In contrast, our work highlights the inherent biases of different models, showing that

various architectures tend to emphasize distinct frequency information. Based on this finding, we develop a heterogenous MoE framework, employing inherent biases of various architectures to capture diverse user behavioral patterns.

**Mixture-of-Experts for recommendation.** Mixture-of-Experts (MoE) has emerged as a powerful paradigm in deep learning, improving supervised learning through a specialized architecture of experts focused on specific tasks [1, 12, 14]. In the realm of recommender systems, MMOE [18] first employs multiple gating mechanisms to dynamically determine the contribution of each expert for various tasks. Based on this framework, various efforts have been made to improve the performance of MoE in recommender systems. For example, SNR [17] enhances the flexibility of parameter sharing with sub-network routing. PLE [29] analyzes the seesaw phenomenon in the multi-task learning scenario, and further designs the progressive layered extraction module to alleviate this. Nevertheless, most existing MoE-based studies use MoE primarily as a multi-task solver. In contrast, our work leverages MoE to achieve fine-grained, customized user modeling by assigning specific experts to focus on particular user behavior patterns. Additionally, current approaches often use the same architecture for all experts (*e.g.,* an MLP module)[15, 29]. Although M3[30] recently introduced a mixture of heterogeneous models to capture diverse temporal ranges, it lacks experimental validation and merely assumes that different encoders naturally attend to different temporal patterns. In contrast, our work conducts in-depth experiments, demonstrating that distinct architectures can capture unique frequency components. Based on this, we propose a heterogeneous MoE framework, activating tailored experts for individual users and providing personalized recommendations. Moreover, existing studies often train the model implicitly, leading to expert conflicts and load imbalances. To tackle these issues, we introduce two contrastive learning tasks and a load-balancing auxiliary task, enhancing expert specialization and alignment, and improving training stability.

## 6 CONCLUSION

In this paper, we found that existing sequential encoders have intrinsic architectural biases and tend to focus on specific frequency ranges of user behavior sequences, limiting their ability to capture user diverse behavioral patterns. To overcome this, we proposed FamouSRec, a frequency-augmented mixture-of-heterogeneous-experts framework for personalized recommendations. The MoE framework enables our approach to integrate the strengths of various experts, thus achieving diversified user modeling. Specifically, different from existing methods that employ experts with the same architecture, our framework uses heterogeneous experts, leveraging their inherent biases to encode a wider range of frequency information. We further introduce a frequency-augmented router to select the most appropriate experts for serving each individual. Additionally, two novel contrastive tasks are designed to enhance expert specialization and alignment. Based on these methods, our framework can activate customized experts to encode user preferred behavioral patterns, thus providing personalized recommendations.

For future work, we will consider developing the MoE framework with more types of experts and leveraging their architectural biases to improve frequency domain learning.

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

# A APPENDIX

## A.1 Fourier Transform

*A.1.1 Discrete Fourier Transform.* The Discrete Fourier Transform (DFT) is a widely used computational method in signal processing. Since the input data for sequential recommendation are one-dimensional sequences, we only consider the 1D DFT. Specifically, for a sequence of $\{x_n\}_{n=1}^{N}$, the 1D DFT transforms the original sequence into a sequence of complex numbers in the frequency domain using the formula:

$$X_k = \sum_{n=1}^{N} x_n W_N^{nk}, \quad 1 \le k \le N \tag{18}$$

where $N$ is the length of the sequence, $W_N^{nk}$ is the twiddle factor, and $X_k$ is a complex number representing the signal with frequency $\omega_k = \frac{2\pi k}{N}$. Through this equation, the DFT decomposes a sequence of values into components of different frequencies. The

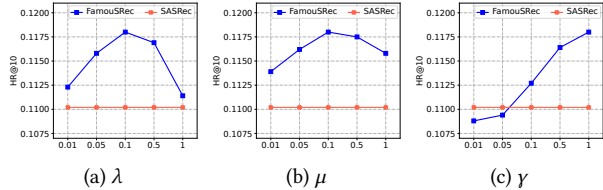

(a) $\lambda$       (b) $\mu$       (c) $\gamma$

**Figure 6: Performance comparison w.r.t. different hyperparameters on the Instruments dataset.**

DFT provides a unique, one-to-one mapping between the time and frequency domains. The frequency representation $\{X_k\}_{k=1}^{N}$ can be transformed back to the original feature domain using the inverse DFT (IDFT), which is given by:

$$x_n = \frac{1}{N} \sum_{k=1}^{N} X_k W_N^{-nk} \tag{19}$$

*A.1.2 Fast Fourier Transform.* To compute the DFT, the Fast Fourier Transform (FFT) algorithm is commonly employed, which breaks down the DFT of a sequence of length $n$ recursively, reducing the time complexity to $O(n \log n)$. The inverse DFT, shown in Equation (19) and structurally similar to the DFT, can also be efficiently calculated using the inverse FFT. In this paper, we employ $\mathcal{F}$ and $\mathcal{F}^{-1}$ to denote FFT and IFFT, respectively.

## A.2 Complexity Analysis

In this part, we analyze the complexity of our proposed FamouS-Rec. Overall, the time complexity of our approach during training primarily arises from four components: frequency domain learning, mixture-of-experts, expert-frequency contrastive learning, and expert alignment contrastive learning. Specifically, for frequency domain learning, we perform FFT and IFFT operations and require a cost of $O(nd \log n)$, where $n$ is the sequence length and $d$ is the hidden size. To use the mixture-of-experts framework for encoding user behavioral sequences, the time complexity is $O(KM(n, d))$, where $K$ is the number of experts and $M(n, d)$ represents the complexity of different model architectures (with Self-Attention at $O(n^2 d)$, GRU at $O(nd^2)$, MLP at $O(nd^2)$, and Mamba at $O(nd + nd \log n)$). In expert-frequency contrastive learning, we employ different experts to encode distinct frequency components, resulting in $K \times K$ encoding operations and a cost of $O(K^2 M(n, d))$. For expert alignment contrastive learning, contrastive learning is conducted for each expert using the output of activated experts as positive samples and other in-batch data as negatives, incurring a cost of $O(B^2 nd + BKM(n, d))$, where $B$ is the batch size. During inference, since we use a sparsely-gated MoE framework that activates only the top-$k$ experts for processing, the time complexity is $O(nd \log n + kM(n, d))k$.

## A.3 Hyper-parameter Analysis

In this section, we examine the effects of key hyperparameters, including the weight hyperparameters $\lambda$, $\mu$, and $\gamma$. To isolate their impact, we adjust one hyperparameter at a time while keeping the others at their optimal values.

**Table 4: Further analysis of the effectiveness of frequency domain modeling. "low-frequency" refers to recommendation results for users with low-frequency behavior patterns.**

| Variants | Instruments | | Arts | |
|---|---|---|---|---|
| | HR@10 | NDCG@10 | HR@10 | NDCG@10 |
| SASRec$_{average}$ | 0.1102 | 0.0652 | 0.1053 | 0.0613 |
| FamouSRec$_{average}$ | 0.1180 | 0.0869 | 0.1103 | 0.0751 |
| SASRec$_{low-frequency}$ | 0.1146 | 0.0679 | 0.1081 | 0.0630 |
| FamouSRec$_{low-frequency}$ | 0.1191 | 0.0786 | 0.1144 | 0.0726 |

Overall, as shown in Figure 6, our approach can outperform the baseline by a significant margin in almost all settings, demonstrating its robustness. In particular, $\lambda$ and $\mu$ are two hyperparameters that control the weight of two contrastive learning tasks, *i.e.,* expert-frequency contrastive learning and expert alignment contrastive learning. By adjusting these values, we observe that selecting appropriate hyperparameter values greatly enhances performance, while values that are either too high or too low can negatively impact it. Furthermore, tuning the load balancing weight $\gamma$ results in noticeable performance shifts, underscoring the importance of this task in ensuring all experts are activated evenly. This task can further help maintain training stability.

## A.4 Analysis of Frequency Domain Learning

Based on the findings proposed in Section 2, we propose the heterogeneous MoE framework. In this part, we delve into its effectiveness in frequency domain learning. Specifically, we take users with low-frequency behavior patterns (referred to as "low-frequency users" for brevity) and SASRec (which is good at capturing low-frequency signals) as an example to analyze the following three questions: 1. Does SASRec perform better for users with low-frequency behavior patterns? 2. Can most users with low-frequency behavior be routed to the self-attention module in our framework? and 3. Can our model provide better recommendations for users with low-frequency behavior compared to SASRec? We conduct experiments in the Instruments and Arts datasets, with the results summarized in Table 4.

For the first question, we compare the average recommendation results of SASRec for all users with those for the low-frequency users. The results show that SASRec provides better recommendations for users with low-frequency patterns, supporting the conclusion that SASRec functions as an effective low-pass filter.

For the second question, we find that 73% of users with low-frequency patterns in the Instruments dataset and 61% in the Arts dataset are routed to the self-attention module during inference. This indicates that our router can effectively identify user behavior patterns and activate the appropriate experts to provide customized recommendations.

Finally, regarding the third question, our model demonstrates superior performance compared to SASRec for these specific users. This is likely because our experts, trained using two contrastive learning tasks, are better able to capture user behavior patterns and focus on specific frequency components, thereby providing more personalized recommendations.

## A.5   Limitations

In this section, we discuss the limitations of our approach and explore potential directions for future work. Our work builds an MoE recommender with heterogeneous experts, activating suitable experts for serving each individual. Especially, the proposed frequency-augmented router first analyzes frequency components contained in user behavioral sequences and then conducts customized routing based on the informativeness of these components, implementing a user-level routing strategy. In contrast, recent studies in NLP have adopted a token-level routing strategy and achieved impressive results [5, 10]. In future work, we plan to expand our routing strategy to include item or session-level routing, for fine-grained modeling.

Furthermore, these recent NLP studies suggest that using an MoE framework can achieve better scaling laws [1]. However, given the sparse nature of interaction data, it is non-trivial to explore this in recommender systems. We aim to address this in our future work.

