# OpenReview forum: "Frequency-Augmented Mixture-of-Heterogeneous-Experts Framework for Sequential Recommendation"
_ACM.org/TheWebConf/2025/Conference — WWW 2025 Poster_

### Official Review · Reviewer_Axhz · 2024-11-22

**Novelty:** 5
**Technical Quality:** 5

**Review:**

The paper proposes an MoE framework with diverse model structures to leverage various models for capturing information at various frequencies. It further enhances the specialization and alignment of experts through expert-frequency contrastive learning and expert alignment contrastive learning. The extensive  experimental results demonstrate the effectiveness of the proposed method.

Pros:

* The paper is well-organized and easy to understand.
* The empirical studies on user behavior and model architecture in Section 2 are meaningful. Using various model structures to capture information at various frequencies could be beneficial for future research.
* The experimental evaluations are extensive. The author compares FamouSRec with ten baseline models of four different types across five datasets to evaluate performance, demonstrating the method's effectiveness.

Cons:

* Although the author provided the framework for FamouSRec, specific implementation details, such as which models were used as expert models in the experiments and the size of FamouSRec's model parameters, remain unclear.
* Since the author employs different experts to capture various frequency information, but in Section 4.1.4 mentions that the number of experts is only adjusted within the range of {3,4}, the effect of the number of experts on performance remains unclear.
* Based on Section 4.1.4, it seems necessary to manually experiment with different combinations of expert model structures to achieve the optimal combination, which might be inefficient.
* There are some writing issues. For example, the parameter $\tau$ appears in Formula 14 in lines 487-490, but the definition of the parameter $\tau$ as the temperature parameter is given in line 501.

**Questions:**

* Could you provide the implementation details of FamouSRec, such as the model parameter size?
* Have you tried studying the impact of the number of different experts on performance?
* Is there an effective or empirical way to choose which model structures to use as experts?
* The parameters $\tau$ are involved in both Formula 14 and Formula 15, but only the $\tau$ in Formula 15 is defined. Are the two $\tau$ the same?

**Reviewer Confidence:**

3: The reviewer is confident but not certain that the evaluation is correct

**Scope:**

4: The work is relevant to the Web and to the track, and is of broad interest to the community

---

### Official Review · Reviewer_ygAV · 2024-11-28

**Novelty:** 5
**Technical Quality:** 5

**Review:**

Strengths:
1. Writing quality
- solid notations and equations.
- The paper is well-organized.
- matrix shapes are denoted for easy understanding.

2. Well-motivated with empirical studies

Weaknesses:
1. The paper lacks justifications for the architecture design.
- Why is the gating network put after IFFT? not between FFT and IFFT? The gating network should capture the frequency preference.
- Why deploy both multiple frequency filters and multiple encoders? As far as I understand, Figure 1 shows the necessity for utilizing multiple frequency filters. However, why do we need multiple encoders?
- For each encoder $M_j$, the input is $x_j$ which is constructed based on $\\tilde{H}_j$ and $Q_j$. Then, each (learnable) filter is aligned with each encoder?
- Why is the input of gating network $X$, not $M_j (x_j)$?

2. Minor concerns
- where is $\\textit{H}$ in Figure 3?
- consistent between $F_u$ and $f_n$. It would be beneficial to denote the matrix shape. $f_n$ should be $f_u^n$?

**Questions:**

please refer to Weaknesses.

**Reviewer Confidence:**

4: The reviewer is certain that the evaluation is correct and very familiar with the relevant literature

**Scope:**

3: The work is somewhat relevant to the Web and to the track, and is of narrow interest to a sub-community

---

### Official Review · Reviewer_5ZPW · 2024-12-02

**Novelty:** 3
**Technical Quality:** 3

**Review:**

Pros:

-The idea of using different sequence architectures to capture different patterns in sequences is reasonable.

-To capture information at different frequencies, the authors used an MOE (Mixture of Experts) architecture for aggregation.

-The authors conducted extensive experiments to validate their idea.

Cons:

-The paper does not compare with ensemble learning methods.

-Using more experts to improve experimental results is fairly common.

**Questions:**

See the Cons above.

**Ethics Review Flag:**

Yes

**Reviewer Confidence:**

4: The reviewer is certain that the evaluation is correct and very familiar with the relevant literature

**Scope:**

4: The work is relevant to the Web and to the track, and is of broad interest to the community

---

### Official Review · Reviewer_c5v5 · 2024-12-02

**Novelty:** 4
**Technical Quality:** 4

**Review:**

Paper Summary:
This paper presents a heterogeneous expert hybrid framework for frequency domain enhancement. The aim is to design specific experts for different user behavior patterns to address the architectural bias of a single model encoder in sequence recommendation. The authors conducted experiments on five datasets, testing both single and hybrid heterogeneous architectures. They also explored the impact of different model components on the overall model. The results demonstrate that the framework outperforms most models.

Paper Strengths:
1. Thorough experimentation: Testing on five datasets enhances confidence in the results. Various models from the literature are compared, from the traditional GRU4Rec and SASRec to the frequency enhancement methods FMLPRec and FEARec to the latest aggregation methods.
2. The paper is well-written and well-organized.

Paper Weaknesses:
1. The article uses four models as experts, but it does not mention the inherent model bias of MLP and does not draw its spectrogram, which makes it confusing as an expert.
2. The choice of datasets seems somewhat niche, and the classic datasets YELP and ML1M are not used.
3. There is an extra ellipsis between w1 and w2.  Additionally, the MLP model is not drawn in the overall framework of the model drawing.

**Questions:**

1. How does the inherent bias of the MLP model influence its role as an expert, and could the inclusion of its spectrogram enhance clarity and justify its selection?
2. What criteria were considered in selecting the datasets, and why were classic datasets like YELP and ML1M excluded from the evaluation?

**Reviewer Confidence:**

3: The reviewer is confident but not certain that the evaluation is correct

**Scope:**

3: The work is somewhat relevant to the Web and to the track, and is of narrow interest to a sub-community

---

### Official Review · Reviewer_XFVL · 2024-12-02

**Novelty:** 5
**Technical Quality:** 5

**Review:**

**Paper Summary**

This paper proposes FamouSRec (an abbreviation of the title), which uses a Mixture-of-Experts (MoE) approach, allowing the model to focus on different frequency ranges via heterogeneous encoder modules. FamouSRec optimizes the model with four losses by utilizing several types of encoder modules (self-attention, mamba, GRU, etc.), each designed to focus on distinct frequency ranges. The model uses the output of MoE through sparse gating to learn CE loss with two contrastive losses, Expert-Frequency Contrastive Loss and Expert Alignment Contrastive Loss and additionally includes a loss that considers load balancing. Validated the model performance through various experiments.

**Summary of Strengths**

S1. The limitation that previous models often exhibit architectural biases, which lead them to focus on specific frequency ranges, is effectively addressed in the proposed approach using an MoE framework.

S2. An ablation study demonstrates that the implemented contrastive loss and gate balancing loss enhance model performance.

S3. The Homogeneous Expert and Discrete Routing experiment further highlights the advantages of Heterogeneous Experts and Customized Routing.

**Summary of Weaknesses**

W1. The ablation study shows that when Frequency Domain Learning is excluded, Recall@10 performance drops significantly, even lower than the Recall@10 performance of SASRec alone. This suggests that only using MoE without a learnable frequency filter doesn’t have enough gain.

W2. A critical point is that there are no experiments demonstrating the direct impact of the MoE-based model on handling multiple frequency ranges. While the performance improves with the MoE framework, it remains unclear whether this improvement is specifically due to better frequency coverage or the effect of combining multiple models in an ensemble.

W3. Apart from the ablation study, there are fewer analytical experiments, making the evaluation insufficient. Additionally, it would strengthen the paper if it’s compared to BSARec[1] or SLIME4Rec[2], which is a state-of-the-art frequency-based model.

**Comments Suggestions and Typos**

C1. In Figure 2, the visualization of frequency attention shows that each model captures a different frequency range, but further explanation on how amplitude values were extracted for each model would be helpful.

C2. In Appendix A.4, only results on low-frequency behavior patterns are shown, while it would be beneficial also to see results for high-frequency and band-pass frequency behavior patterns.

C3. Equation (10): Typo correction needed - "Softamx" should be "Softmax."

[1] Shin, Yehjin, et al. "An attentive inductive bias for sequential recommendation beyond the self-attention." Proceedings of the AAAI Conference on Artificial Intelligence. Vol. 38. No. 8. 2024.
[2] Du, Xinyu, et al. "Contrastive enhanced slide filter mixer for sequential recommendation." 2023 IEEE 39th International Conference on Data Engineering (ICDE). IEEE, 2023.

**Questions:**

Q1. How did you extract the frequency amplitude of the filter model? It is wondering if filter models are not trained under the frequency domain.

**Reviewer Confidence:**

4: The reviewer is certain that the evaluation is correct and very familiar with the relevant literature

**Scope:**

3: The work is somewhat relevant to the Web and to the track, and is of narrow interest to a sub-community